# Combustion Synthesis of High Density ZrN/ZrSi_2_ Composite: Influence of ZrO_2_ Addition on the Microstructure and Mechanical Properties

**DOI:** 10.3390/ma15051698

**Published:** 2022-02-24

**Authors:** Zaki I. Zaki, Saad H. Alotaibi, Bashayer A. Alhejji, Naser Y. Mostafa, Mohammed A. Amin, Mohsen M. Qhatani

**Affiliations:** 1Department of Chemistry, Faculty of Science, Taif University, Taif 21974, Saudi Arabia; b.o@live.no (B.A.A.); nmost69@yahoo.com (N.Y.M.); maaismail@yahoo.com (M.A.A.); mohsen@tu.edu.sa (M.M.Q.); 2Department of Chemistry, Faculty of Turabah University College, Taif University, Turabah 21995, Saudi Arabia; s.alosaimi@tu.edu.sa

**Keywords:** zirconium nitride, composites, hardness, synthesis, microstructure

## Abstract

In this study, a high-density ZrN/ZrSi_2_ composite reinforced with ZrO_2_ as an inert phase was synthesized under vacuum starting with a Zr-Si_4_N_3_-ZrO_2_ blend using combustion-synthesis methodology accompanied by compaction. The effects of ZrO_2_ additions (10–30 wt%) and compression loads (117–327 MPa) on the microstructure, porosity and hardness of the samples were studied. The process was monitored using XRD, SEM, EDS, porosity, density and hardness measurements. Thermodynamic calculations of the effect of ZrO_2_ addition on the combustion reaction were performed including the calculation of the adiabatic temperatures and the estimation of the fractions of the liquid phase. The addition of up to 20 wt% ZrO_2_ improved the hardness and reduced the porosity of the samples. Using 20 wt% ZrO_2_, the sample porosity was reduced to 1.66 vol%, and the sample hardness was improved to 1165 ± 40.5 HV at 234 MPa.

## 1. Introduction

The transition metal nitrides group is related to a family of materials featuring a combination of exceptional characteristics, such as a high melting point, very high hardness, metallic luster, and sometimes shining colors. Moreover, their simple metallic structures are usually associated with outstanding thermal and electrical conductivities [1]. Owing to its unique physical and chemical properties, zirconium nitride ZrN is considered an important candidate of the nitrides group. Together with the nitrides of Ti and Hf, ZrN can melt at ambient pressure without decomposition. ZrN has a small coefficient of thermal expansion (7.2 × 10^−6^ K^−1^) and a high coefficient of thermal conductivity (21.9 Wm^−1^ K^−1^) and very small resistivity 13.6–24.0 μΩ cm at 300 K [1,2,3,4,5,6].

ZrN-based materials have been used in various fields of industrial and nuclear applications. ZrN is generally used as a refractory slag-resistance composite [7] and refractory conductor. It has some applications in the microelectronics industry as a diffusion barrier [8] and Josephson junction [9]. ZrN is widely used as hard coating for cutting tools [10,11], decorative coatings, heat-mirrors [12], and solar control coatings [13]. Moreover, ZrN is used as an inert matrix fuel (IMF) for the transmutation of actinides [14,15,16]. For use as IMF, ZrN materials must possess exceptional anti-irradiation properties; consequently, highly dense ZrN materials are of essential interest [17]. ZrN has been examined as a nuclear fuel in nitride fuels, such as (Pu, Zr)N and (U, Zr)N. Research has shown that mixed nitride fuels containing ZrN, exhibit a better swelling attitude and reduce the interaction between nitride fuel and cladding [18].

ZrN powders can be densified using different sintering methodologies including pressureless sintering, hot pressing, isostatic hot pressing, and spark-plasma sintering [19,20]. The sintering techniques of ZrN usually require a source of very high temperatures of up to 2000 °C, long periods and/or the application of high compression loads. The difficulties of ZrN sintering arise from its low self-diffusion coefficient, high melting temperature, and the presence of an oxide layer on the surfaces of ZrN grains [21,22]. Reducing sintering temperature is an essential subject for the commercial production of high-density ZrN. The formation of a liquid phase of low-melting-point metals or ceramics has been investigated by many authors. Zr and Ti additives have been reported to assist in ZrN sintering. The addition of approximately 20 mol% Ti or Zr to ZrN led to more than 98% relative densities by hot pressing at 1700 °C compared with about 2000 °C for sintering of pure ZrN [22]. However, the precipitation of the metallic phase in the grain boundaries of ZrN reduced its mechanical properties at high temperatures, altered its electrical and thermal behaviours and changed its anti-irradiation performance [23]. On the other hand, ceramic-sintering aids are considered to be a significant approach for improving the sintering behaviour of ceramics. Various sintering aides including ZrSi_2_, Si_3_N_4_, MoSi_2,_ SiC, and AlN have been found to significantly enhance the sintering of ceramics through the formation of liquid phases and the generation of fine microstructures [24,25,26].

Additionally, ZrSi_2_ is reported to be an efficient sintering aid for ZrN [27] and ZrB_2_ [24,26,28,29,30]. Furthermore, SiC/ZrSi_2_ mixtures have been reported to be suitable sintering aids for ZrB_2_ ceramics because they enhance their sintering and mechanical properties [26,31,32]. Nearly fully dense zirconium diboride ceramics were synthesized using 5 wt% SiC platelets and 5 wt% zirconium silicide via hot pressing at 1800 °C for 60 min in an argon atmosphere [26]. Shu-Qi Guo et al. prepared fully dense ZrB_2_-based composites using 20 vol% ZrSi_2_ by sintering at 1650 °C for 60 min under vacuum [29]. High-density monolithic ZrN has been fabricated without sintering aids via spark plasma sintering using commercially available ZrN powder [33]. The spark-plasma-sintering technique, with its lower temperature and time, increased the relative density of ZrN to only 89.1% at 1800 °C [19].

On the other hand, ZrN/ZrO_2_ composites are rarely mentioned in the literature, and ZrO_2_ is commonly used as an additive for increasing the fracture toughness of ceramics. This effect is based on the martensitic (tetragonal to monoclinic) phase transformation of ZrO_2_, accompanied by a volume increase in the order of 3–6%. This increase in volume generates stresses inside the ceramic matrix and obstructs crack propagation [34].

In this study, a composite of ZrN/ZrSi_2_/ZrO_2_ will be synthesized from Zr, Si_3_N_4_, and ZrO_2_ powders using combustion synthesis under load methodology [35,36] to obtain dense objects. The effects of ZrO_2_ as an inert reinforcement and the application of different compression loads will be investigated. To the best of our knowledge, no available studies reported the fabrication of ZrN/ZrSi_2_/ZrO_2_ composites by combustion synthesis.

## 2. Materials and Methods

### 2.1. Materials

This study used silicon nitride Si_3_N_4_ powder (mean particle size 1.45 µm & 99.0% purity, Atlantic comp., Bergenfield, NJ, USA), zirconium metal powder Zr (mean particle size 14.3 µm & 98.0% purity Riedel, USA), and zirconia powder ZrO_2_ (99% purity and mean particle size 8.03 µm, Sigma-Aldrich, St. Louis, MO, USA).

### 2.2. Procedure

The required weights of the reaction mixture were calculated depending on the reaction (1).
11Zr + 2Si_3_N_4_ + xZrO_2_ = 3ZrSi_2_ + 8ZrN + xZrO_2_(1)

Different mole fractions of ZrO_2_ were investigated; 1.15, 2.60 and 4.46, corresponding to weight fractions of 10, 20, and 30 wt%, respectively. Weighed amounts of Zr, Si_3_N_4_ and ZrO_2_ were mixed in an agate mortar for 20 min. The powder mixtures were compacted at 60 MPa into green cylindrical shapes (1.0 cm height and 1.20 cm diameter) using a manual hydraulic pressing machine. The compact was then surrounded by a resistor coil and sited inside a steel mold (φ 5 cm × 10 cm). Sand powder was poured into the die and covered the green compact from all directions, and then the reactor was evacuated. Sample ignition was accomplished by increasing the temperature of the whole sample using the resistor coil. A mechanical load was applied during the reaction using a motorized hydrolytic press on the steel die. A type-C thermocouple was used to monitor the temperature of the compact [37]. After the reaction, the sample was removed and cut from its circular face into two pieces using a precision cutting machine. Using this procedure, the main parameters affecting the synthesis process were examined. These parameters included ZrO_2_ weight fractions (10–30 wt%), pressing load (117–327 MPa) and delay time.

### 2.3. Characterization

The sample phases were investigated using an X-ray diffractometer (D8 Advanced Bruker AXS, GMbH, Karlsruhe, Germany). The sample microstructures were examined using scanning electron microscope SEM (JSM-5410, JEOL, Tokyo, Japan). The compositions of different phases were determined using electron dispersive spectroscopy (EDS). The samples were mounted using epoxy resin, polished, and coated with gold prior to SEM examination. The hardness of the specimens was tested using an LV 800 AT digital display microhardness tester according to ASTM C1327-15 [38]. The porosity and apparent density of the samples were measured using Archimedes’ method [39,40]. Archimedes’ principle is based on saturating the pores of a sample with water by immersing it in water and boiling for 2 h.

## 3. Results and Discussion

### 3.1. Thermodynamic Study of the ZrN-ZrSi_2_ with xZrO_2_ Additions

A thermodynamic study of the ZrN-ZrSi_2_/ZrO_2_ system was conducted based on reaction (2) and Equation (3):(2)$$11Zr+2Si_{3}N_{4}+xZrO_{2}=3ZrSi_{2}+8ZrN+xZrO_{2}\;\;\;\;\;\;\;\;\;\Delta H_{r,298}=-1953.46\;kJ$$
(3)$$\begin{array}{cc} -\Delta H_{r,298}+\;\int_{298}^{T_{o}} & C_{p}{\left(11Zr+2Si_{3}N_{4}+xZrO_{2}\right)}_{s}dT\;\\ & =8\int_{T_{o}}^{3225}C_{p}{\left(ZrN\right)}_{s}dT\;+\;8\alpha \int_{3225}^{T_{ad}}C_{p}{\left(ZrN\right)}_{l}dT\;\\ & +\;3\int_{T_{o}}^{1893}C_{p}{\left(ZrSi_{2}\right)}_{s}dT\;\\ & +\;3\beta \int_{1893}^{T_{ad}}C_{p}{\left(ZrSi_{2}\right)}_{l}dT\;+x\int_{T_{o}}^{2950}C_{p}{\left(ZrO_{2}\right)}_{s}dT\;\\ & +\;x\gamma \int_{2950}^{T_{ad}}C_{p}{\left(ZrO_{2}\right)}_{l}dT\;+8\alpha \Delta H_{f,ZrN}+3\beta \Delta H_{f,ZrSi_{2}}\\ & +\;x\gamma \Delta H_{f,ZrO_{2}} \end{array}$$
where ${\Delta H}_{r,298}$ is the enthalpy of reaction (2) at 298 K, *C*_*p*_ is the heat capacity, *T*_*o*_ is the initial temperature, *T*_*ad*_ is the adiabatic temperature, $\Delta H_{f}$ is the enthalpy of fusion, x is the number of moles of ZrO_2_. α, β and γ represent the values of the molten fractions of ZrN, ZrSi_2_ and ZrO_2_, respectively. The total liquid phase was calculated depending on the values of α, β and γ.

It is worth mentioning that the calculations were carried out at the measured ignition temperature (1241 K) based on the assumption that ZrO_2_ is an inert phase and does not participate in reaction (2). Figure 1 represents the effect of ZrO_2_ addition on the adiabatic temperature of reaction (2) and the molten fractions of ZrSi_2_, ZrO_2_, and ZrN and the total liquid phase that developed at the measured ignition temperature (1241 K). Generally, the adiabatic temperature, molten fractions of ZrSi_2_, ZrO_2_, and ZrN, and the total liquid phase decrease with increasing amounts of ZrO_2_ in the reaction mixtures owing to the dilution effect of ZrO_2_ addition. The fraction of molten ZrSi_2_ was not affected by ZrO_2_ addition and remained constant at 100 wt%. On the other hand, the fraction of molten ZrN liberated at 1241 K decreased from 28.7 wt% in the case of no addition, to 0 wt% in case of 10–30 wt% ZrO_2_ additions. Because of the lower melting point of ZrO_2_ (2950 K) compared to that of ZrN (3225 K), ZrO_2_ melted instead of ZrN. The addition of ZrO_2_ as an inert phase (diluent) absorbs some of the energy liberated from the combustion reaction for its heating and its physical change from the solid to the liquid phase. This resulted in a drop in the calculated *T_ad_* from 3225 to 2788 K with increasing ZrO_2_ addition from 0 wt% (x = 0) to 30 wt% (x = 4.47 moles), with a corresponding decrease in the total liquid phase from 53.3 to 24.1 wt% of the sample, respectively.

In order to understand the different physical changes of the product and relate them to the heat given to the reactants and the heat released during the reaction, comprehensive thermodynamic calculations were carried out using 20 wt% of the ZrO_2_-sample as an example, Figure 2. Heating the reaction mixture of a 20 wt% ZrO_2_-sample to the ignition temperature (1241 K) should supply the system with 796.8 kJ, while the reaction itself provides the system with 1982.9 kJ. The calculated total liquid phase for the 20 wt%-ZrO_2_ sample was 32.19 wt% with an adiabatic temperature equal to the ZrO_2_ melting point (2950 K).

It is expected that, under adiabatic conditions, the reaction product should preserve the same amount of heat (796.8 kJ) given to the reactants and, accordingly, the same temperature (1241 K). However, the calculation using 796.8 kJ shows that the temperature of the product reaches 1280 K, exceeding the reactant temperature (1241 K). The main reason for this phenomenon is the difference between the heat capacity values of the products and that of the reactants. The heat capacity change of reaction-1 is equal to 17.2 J/k which means that the sum of the heat capacity of the products is less than that of the reactants. This should contribute to a logical increment in the product temperature above the reactant temperature. Another probable reason is the phase transformation of the zirconium content of the reactants at 1135 K. At atmospheric pressure and temperatures of up to 1135 K, zirconium transforms from a hexagonal close-packed structure (α-phase) into a body-centered cubic structure (β-phase) [41]. This consumes 4.01 kJ per mole of zirconium with a total of 44.14 kJ out of 796.8 kJ on the expense of raising the reactant temperature. This amount of latent heat of transformation also contributes to a decrease in the reactants’ temperature.

The temperature of the product will then increase from 1280 K to the melting point of ZrSi_2_ (1893 K), expending 577.9 kJ. The temperature remains constant at 1893 K until the complete conversion of zirconium silicide to the molten state with a consumption of 234.5 kJ. The temperature of the product will begin to rise again to 2950 K (ZrO_2_ melting point) utilizing 1117.3 kJ. The product temperature should be stable at 2950 K until the entire amount of ZrO_2_ has changed to liquid phase. However, the rest of the system heat content (53.2 kJ) is sufficient only to melt 23.51 wt% of ZrO_2_ (0.61 mol).

### 3.2. Effect of ZrO_2_ Weight Ratios

In this series of experiments, different weight percentages of ZrO_2_ ranging from 10 to 30 wt% were used in a trial to improve the composite properties. The samples were prepared using 14.23 µm Zr, and 8.03 µm ZrO_2_ at a compression load of 234 MPa. All samples were successfully ignited in the explosive mode of combustion. To the naked eye, the samples had high densities with metallic lusters.

An XRD diagram of the samples with and without ZrO_2_ additives from which the crystallite sizes and the Miller indices have been calculated are provided in Figure 3. The diffraction patterns of the different samples showed the appearance of ZrN and ZrSi_2_ phases as the main phases. The intensity of the ZrO_2_ phase appeared as a minor phase and increased with increasing amounts in the reaction medium. The appearance of ZrN and ZrSi_2_ phases besides ZrO_2_ phases ensured that the presence of ZrO_2_ during the combustion process did not retard the main reaction between Zr and Si_3_N_4_, and worked as an inert phase without participating in the reaction.

Porosity and density measurements of the samples containing different ZrO_2_ weight ratios are given in Figure 4. For up to 20 wt% ZrO_2_ addition, a noticeable decrease in the sample porosity was recorded while the bulk densities had almost the same value close to the apparent density values. The 10 wt% ZrO_2_-sample reached a minimum porosity value (less than 1.0 vol%), with a maximum bulk density value. The porosity decreased from 2.7 in the case of no additions to 1.66 vol% at 20 wt% ZrO_2_. Increasing the ZrO_2_ fraction by more than 20 wt% led to a significant increment in the sample porosity up to 15.2 vol% at 30 wt% ZrO_2_ additives. The increase in the samples’ porosity by increasing the amount of ZrO_2_ additives can be explained depending on the data obtained from the thermodynamic calculations of the systems having different ZrO_2_ contents, shown in Figure 1. It could be noticed that additions of ZrO_2_ led to a gradual decrease in the adiabatic temperature and the fraction of the total liquid phase. It is well known that the amount of liquid phase strongly controls the sample sintering [22,36]. Figure 5 simply shows that the increase in sample porosities with ZrO_2_ addition was strongly related to the decrease in both the total liquid phase and adiabatic temperatures.

On the other hand, it was noticed that the measured apparent densities of the samples were a little bigger than the samples’ theoretical densities, which is not the usual case. This meant that the actual product had a higher density than the targeted one, which may suggest the formation of other phases besides the planned ones. For this reason, an SEM investigation and EDX analysis of the different products were carried out as illustrated in Figure 6.

Generally, the microstructure was carried out using the back scattered electron mode BSE, Figure 6. It should be stated that the brightness of the phase in the BSE image depends on its heaviness such that the heavier the phase is, the brighter its appearance in the image. The samples’ microstructure could be differentiated into three main areas. The first area was the matrix (dark gray) which was similar in its composition to ZrSi_2_ phase. The second area was the bright particles which mainly composed of Zr-N. The last one is the light gray phase areas which were composed mainly of Zr and Si with silicon deficiency. The formation of Zr-Si system with a lower Si content than that of ZrSi_2_ should contribute to increasing the samples’ apparent densities over the theoretical ones calculated under the assumption of only ZrSi_2_ formation. On the other hand, microstructure investigations showed that the bright particles mainly ZrN had almost no sintering and that only the Zr-Si matrix was responsible for sample’ sintering. This aligned well with thermodynamic calculations which predicted that ZrN phase would be liberated in the solid state and only the ZrSi_2_ phase would be completely melted.

The hardness measurements of the samples reinforced with ZrO_2_ additives are shown in Figure 7. There was a gradual increase in the hardness of the samples with an increasing ZrO_2_ content up to 20 wt%. The sample hardness achieved a maximum value (1165 ± 40.5 HV) at 20 wt% ZrO_2_. The further addition of ZrO_2_ had a negative impact on the sample’ hardness, with a significant drop to 747 ± 61.2 HV at 30 wt% ZrO_2_. The drop of the hardness at 30 wt% ZrO_2_ could be correlated with the sudden increase in the sample porosity, as shown in Figure 7. Both the increased sample porosity and the decreased value of hardness arose from a decrease in the adiabatic temperature and the liquid phase content as predicted by the thermodynamic calculations, Figure 1. Although the 20 wt% ZrO_2_ sample had a higher porosity than the 10 wt% ZrO_2_ sample, it had the highest hardness. For samples with 10 and 20 wt ZrO_2_, two factors controlled the hardness, that is, the porosity and the amount of ZrO_2_. The porosity tended to decrease the hardness while the ZrO_2_ tended to improve its mechanical properties. The porosity increased to only 1.66 vol% in case of 20 wt% ZrO_2_ and was considered to be very low. Therefore, the improvement in mechanical properties due to 20 wt% was greater than the increase in porosity.

Regarding the effect of crystallite sizes on the samples’ hardness, it was reported that the hardness increased with a reduction in the crystallite size [42]. With respect to this work, it was clear from Table 1 that the crystallite sizes of the different phases were almost the same and were not affected by the addition of ZrO_2_. According to the hardness measurements and microscopic investigations, the work continued using the sample reinforced with 20 wt% of ZrO_2_.

### 3.3. Effect of Compression Load on ZrN/ZrSi_2_ Composite Having 20 wt% ZrO_2_

This series of experiments was devoted to a study of the effect of applying different compression loads on the sample during the combustion process. Compression loads ranging from zero up to 327 MPa were investigated with respect to the porosity and hardness of the samples. The effects of different compression loads on the samples’ porosity, bulk and apparent densities are illustrated in Figure 8. From this figure, it could be seen that the sample obtained without compression had too high a porosity of approximately 47 vol%. When applying only 117 MPa on the sample during combustion, the porosity decreased significantly to ~14 vol% and the samples’ bulk densities approached their apparent values. The sample porosity reached less than 1.0 vol% when 327 MPa was used during the combustion reaction.

The samples’ hardness was measured using a hardness tester and the results are given in Figure 9. The sample obtained without pressing had a very weak structure and its hardness could not be measured. With an increased pressing load, the sample hardness consecutively improved to 1165 ± 40.5 HV at 234 MPa. The hardness of the sample decreased slightly to 1121 ± 8.1 HV by increasing the load to 327 MPa.

### 3.4. Delay Time

The timing of the compression load application (327 MPa) was delayed for only 2 s after ignition, to provide an opportunity for degassing the sample. However, there was a dramatic decrease in the sample hardness and a significant increase in the sample porosity upon delaying the timing of compression for only 2 s. The sample hardness decreased from 1121 ± 8.1 to 567 ± 31.2 HV, accompanied by an increase in the sample porosity from less than 1.0 to 13.33 vol%. This was because during these 2 s, the temperature of the sample greatly decreased due to the rapid heat dissipation of the combustion reactions.

## 4. Conclusions

Combustion synthesis under load methodology has been used successfully to produce high-density ZrN/ZrSi_2_/ZrO_2_ composites. An increase in the ZrO_2_ content and in the compression load to 20 wt% and 324 MPa, respectively was found to have a positive impact on the hardness and porosity of the samples. The sample hardness was improved to 1165 ± 40.5 HV and the sample porosity was reduced to 1.66 vol% upon the addition of 20 wt% ZrO_2_. A further increase in the ZrO_2_ content led to a sudden drop in hardness with a noticeable increase in the sample porosity. The optimum compression load was 234 MPa, and below or above this value, the sample hardness decreased.

Thermodynamic calculations indicated that 100% of ZrSi_2_ would be in the molten state while the entire amount of ZrN would be in the solid state. This was well aligned with the SEM images, which showed ZrSi_2_ as a continuous phase and ZrN particles disseminated without sintering and cemented only by the ZrSi_2_ matrix.

## Figures and Tables

**Figure 1 materials-15-01698-f001:**
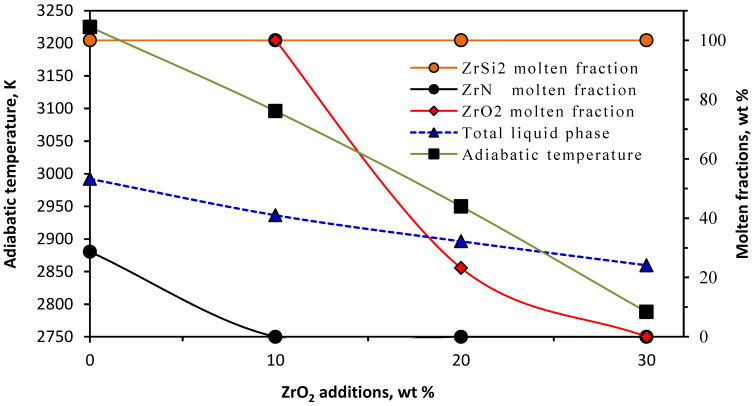
Effect of ZrO_2_ additions on the adiabatic temperature, molten fractions and total liquid phase at the ignition temperature (1241 K).

**Figure 2 materials-15-01698-f002:**
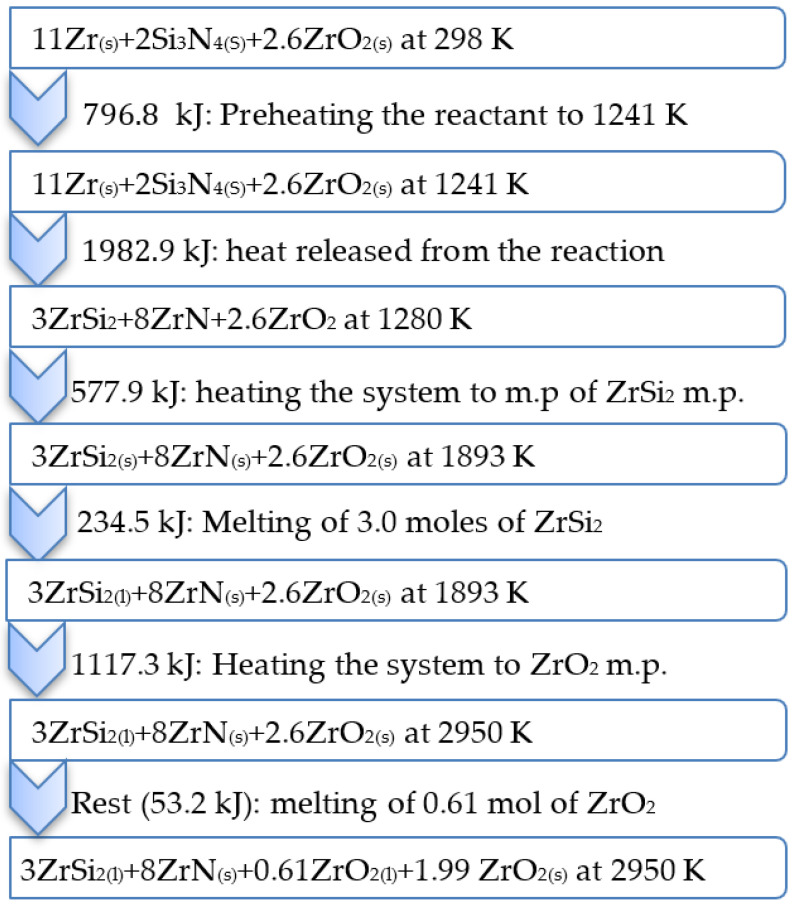
Schematic illustration of the different physical changes of 20 wt% ZrO_2_-sample (2.6 mol ZrO_2_).

**Figure 3 materials-15-01698-f003:**
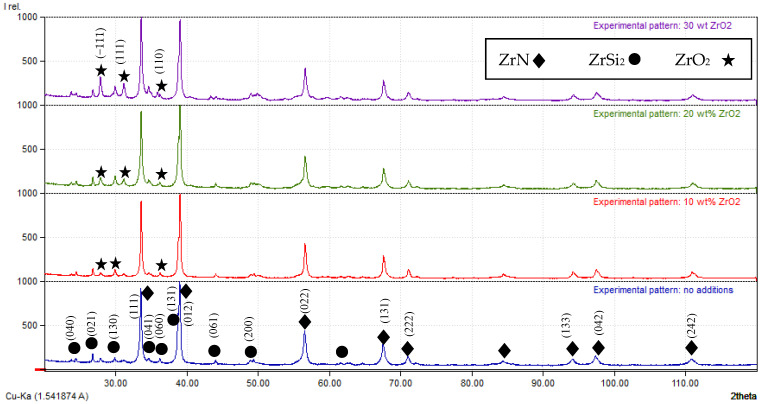
XRD diagrams of the combustion products having different wt% of ZrO_2_ prepared at 234 MPa.

**Figure 4 materials-15-01698-f004:**
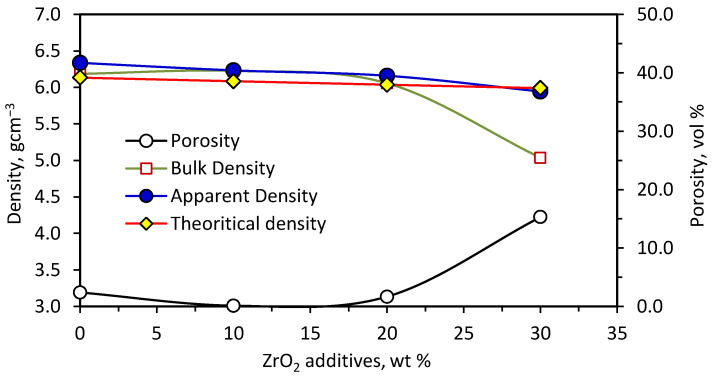
Effect of ZrO_2_ amounts on the ZrN/ZrSi_2_-samples’ porosity, bulk, apparent and theoretical densities (prepared at 234 MPa).

**Figure 5 materials-15-01698-f005:**
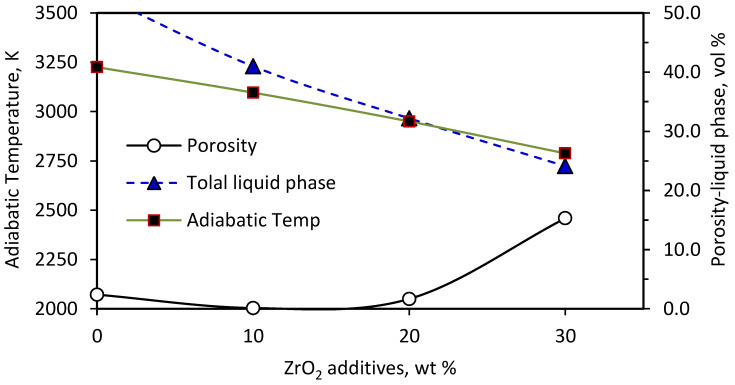
Effect of ZrO_2_ amounts on the ZrN/ZrSi_2_-samples’ porosity prepared at 234 MPa, total liquid phase and adiabatic temperature.

**Figure 6 materials-15-01698-f006:**
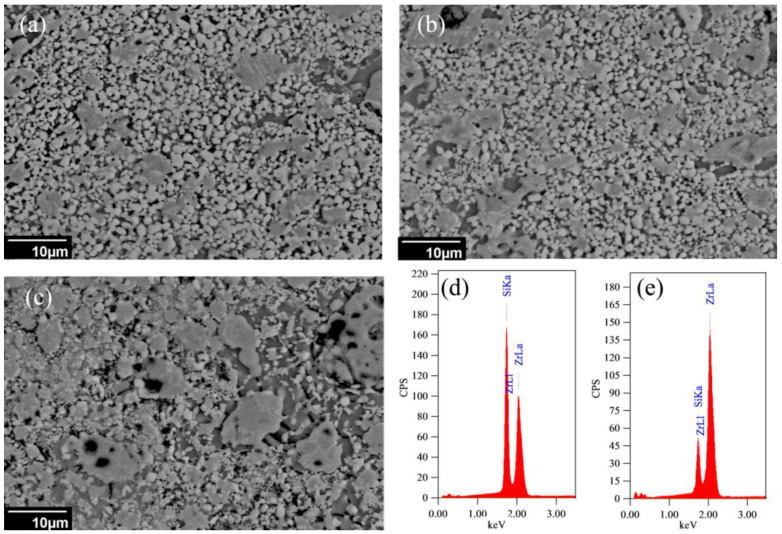
BSE image of ZrN/ZrSi_2_ reinforced with different ZrO_2_ wt% prepared at 234 MPa: (**a**) 10 wt% (**b**) 20 wt% (**c**) 30 wt% (**d**) EDX of dark gray area (sample-b) (**e**) EDX of light gray areas (sample-b).

**Figure 7 materials-15-01698-f007:**
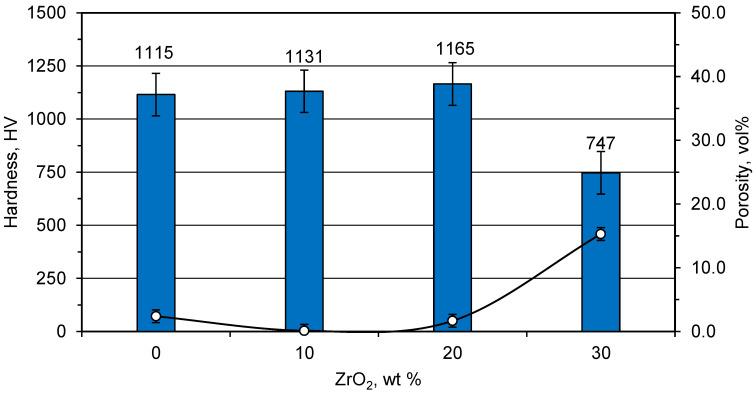
Effect of ZrO_2_ additives on the samples’ hardness (bars) and porosity (data ponits) prepared at 234 MPa.

**Figure 8 materials-15-01698-f008:**
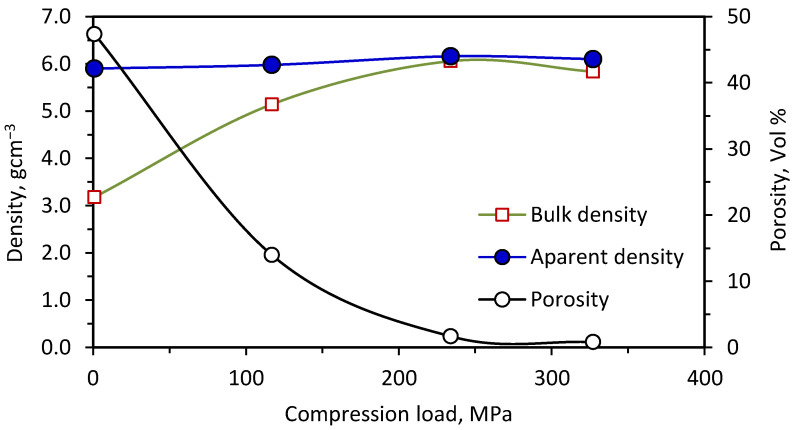
Effect of compression load on the 20 wt% ZrO_2_-samples’ porosity, bulk and apparent densities.

**Figure 9 materials-15-01698-f009:**
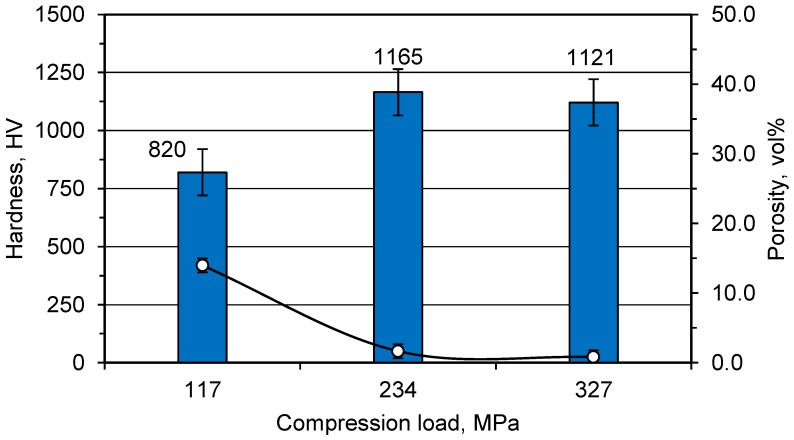
Effect of pressing load on the samples’ hardness (bars) and porosity (data points) reinforced with 20 wt% ZrO_2_.

**Table 1 materials-15-01698-t001:** Crystallite sizes of different phases calculated from the samples’ XRD diagrams.

Phases	Crystallite Size, nm			
	0% ZrO_2_	10% ZrO_2_	20% ZrO_2_	30% ZrO_2_
ZrN	21.0	26.1	20.8	23.8
ZrSi_2_	24.7	22.4	22.2	22.4
ZrO_2_	---	248.6	247.3	245.9

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
