# Peer review of "Combustion Synthesis of High Density ZrN/ZrSi2 Composite: Influence of ZrO2 Addition on the Microstructure and Mechanical Properties"

_materials, 2022, doi:10.3390/ma15051698_

Round 1
Reviewer 1 Report
The article has novelty and is of interest to some materials science brunches since no available literature reported the fabrication of ZrN/ZrSi2/ZrO2 composites by combustion synthesis. It can be accepted for publication in Materials after major revision.
The manuscript requires more detail from the authors.
Comment 1. To a better understanding add the numerical or percentage deviation to the hardness values within the text. The deviation markings in Figure 7 are not enough.
Comment 2. Is it possible to identify the crystallographic planes of phases presented in XRD diagrams for your material? In this case, it'll be possible to characterize preferred orientation in materials that has a direct influence on the properties, namely hardness.
Comment 3. Can you calculate the crystallite size of the experimental materials based on the Sherrer equation (FWHM) evaluated from XRD peaks? The crystallite size also affects the hardness.
Comment 4. Why ZrxOy is not identified on EDX spectra? If it is the top layer in your ZrN/ZrSi2/ZrO2 composites and its weight fractions increases from 10 to 30 %, it undoubtedly will be identified. EDX data should be revised.
Comment 5. On the one hand, the authors write "... the decreased value of hardness were arisen from the decrease in the adiabatic temperature and the liquid phase content as predicted by the thermodynamic calculations, Figure 1." Based on this suggestion, the best hardness should have the sample with the highest adiabatic temperature, i.e. ZrN/ZrSi, but in practice the hardness of this sample has the 3d place. On the other hand, if the hardness is related to the porosity, the best hardness value should have sample ZrN/ZrSi2/ZrO2(10 wt.%), but in practice, the best hardness obtained for ZrN/ZrSi2/ZrO2(20 wt.%). Please, explain.
Comment 6. Within the text, there are some grammar mistakes, which should be corrected.
Author Response
Reviewer 1
Comment 1. To a better understanding add the numerical or percentage deviation to the hardness values within the text. The deviation markings in Figure 7 are not enough.
Answer: The numeric deviations were added through the text.
Comment 2. Is it possible to identify the crystallographic planes of phases presented in XRD diagrams for your material? In this case, it'll be possible to characterize preferred orientation in materials that has a direct influence on the properties, namely hardness.
Answer: The crystallographic planes are identified in the XRD diagram.
Comment 3. Can you calculate the crystallite size of the experimental materials based on the Sherrer equation (FWHM) evaluated from XRD peaks? The crystallite size also affects the hardness.
Answer: A new table was added including the calculated crystallite sizes of the different phases for each sample. The following sentences are also introduced line (320-325).
“Regarding the effect of crystallite sizes on the samples’ hardness it was reported that the hardness increased with a reduction in the crystallite size [41]. With respect to this work it was clear from table 1 that the crystallite sizes of the different phases were almost the same and did not affect by the addition of ZrO2.”
Comment 4. Why ZrxOy is not identified on EDX spectra? If it is the top layer in your ZrN/ZrSi2/ZrO2 composites and its weight fractions increases from 10 to 30 %, it undoubtedly will be identified. EDX data should be revised.
Answer: The EDX analysis is only displayed for the the20wt% ZrO2 sample. I think that the weight percentage of oxygen of this sample (5.1 %) is small and cannot be detected by the EDX.
Comment 5. On the one hand, the authors write "... the decreased value of hardness were arisen from the decrease in the adiabatic temperature and the liquid phase content as predicted by the thermodynamic calculations, Figure 1." Based on this suggestion, the best hardness should have the sample with the highest adiabatic temperature, i.e. ZrN/ZrSi, but in practice the hardness of this sample has the 3d place. On the other hand, if the hardness is related to the porosity, the best hardness value should have sample ZrN/ZrSi2/ZrO2(10 wt.%), but in practice, the best hardness obtained for ZrN/ZrSi2/ZrO2(20 wt.%). Please, explain.
Answer:These are good notes.
Comment: " Based on this suggestion, the best hardness should have the sample with the highest adiabatic temperature, i.e. ZrN/ZrSi, but in practice the hardness of this sample has the 3d place.”
Answer: Although . ZrN/ZrSi has the highest adiabatic temperature, it has no ZrO2. The main conclusion of this work is that the addition of ZrO2 improves the samples’ hardness even if these samples have a lower adiabatic temperature.
Comment:”On the other hand, if the hardness is related to the porosity, the best hardness value should have sample ZrN/ZrSi2/ZrO2(10 wt.%), but in practice, the best hardness obtained for ZrN/ZrSi2/ZrO2(20 wt.%). Please, explain.”
Answer: We should back to the definition of hardness “Hardness is a measure of how difficult or easy it is for a substance to be penetrated or scratched”. It’s clear that the presence of pores will facilitate the material penetration and weaken its structure. And the indenter will go deeper inside the materials. So, there is a logic relation between the porosity and hardness.
For samples having 10 and 20 wt ZrO2, two factors are controlling the hardness i.e. the porosity and the amount of ZrO2. Porosity tends to decrease the hardness while ZrO2 tends to improve the mechanical properties. The porosity increases only to 1.66 vol % in case of 20 wt % ZrO2 and is considered as very low porosity. So, the improvement in mechanical properties due to 20 wt% is greater that the increase in the porosity. Based on this discussion, the following sentences are added to the text line 311 to 318.
“Although the sample having 20 wt % ZrO2 had a higher porosity than the 10 wt % ZrO2 sample, it had the highest hardness. For samples having 10 and 20 wt ZrO2, two factors were controlling the hardness i.e. the porosity and the amount of ZrO2. Porosity tended to decrease the hardness while ZrO2 tended to improve the mechanical properties. The porosity increased only to 1.66 vol % in case of 20 wt % ZrO2 and is considered as a very low porosity. So, the improvement in mechanical properties due to 20 wt% was greater than the increase in the porosity.”
Comment 6. Within the text, there are some grammar mistakes, which should be corrected.
Answer: The grammar was revised by a native English colleague.
Reviewer 2 Report
“Combustion synthesis of high-density ZrN/ZrSi2 composite: Influence of ZrO2 addition on the microstructure and mechanical properties”
This paper reports thermodynamic calculation and the properties of ZrN/ZrSi2/ZrO2 composite by combustion synthesis method. Although of potential interest the study is incomplete and not suitable for publication in its present form. Some items should be considered as follows:
Typos and grammar errors are common in the manuscript. A considerable number of sentences are not comprehensible, some of which can be due to typos or grammar errors. About the manuscript, it is recommended to recheck the grammatical errors or typos.
The state of the art needs to be described more in the introduction. This information could help to show the lack of knowledge in the author's field of research.
In this manuscript, results are well presented, but discussion on the obtained results must be completely provided. Results should be more discussed and compared to the findings from other researchers.
In section 2. Materials and Methods, calculation of mole fractions should be defined clearly.
The standard method for testing microhardness should be provided.
Authors claim reaction 1 was the only reaction could be possibly happened during synthesis. All possible reactions should be provided and compared to each other by thermodynamic software such as HSC.
In Fig.3, some peaks were not identified.
Unlike Fig.5, it seems that the sample was not fully-dense after the synthesis process based on Fig.6. why?
The intrinsic value of hardness for each phase (ZrN,ZrSi2,ZrO2) should be provided to discuss the hardness of the composite by increasing ZrO2 content.
More tests and data are needed to evaluate mechanical properties.
The Conclusion section needs to be rewritten
Author Response
Reviewer 2
1-Typos and grammar errors are common in the manuscript. A considerable number of sentences are not comprehensible, some of which can be due to typos or grammar errors. About the manuscript, it is recommended to recheck the grammatical errors or typos.
Answer: The grammar was revised by a native English colleague.
2-The state of the art needs to be described more in the introduction. This information could help to show the lack of knowledge in the author's field of research.
Answer:The introduction is based on 40 references. I think this number is a quit enough.
2-In this manuscript, results are well presented, but discussion on the obtained results must be completely provided. Results should be more discussed and compared to the findings from other researchers.
Answer:
The results are discussed and compared with 4 references 40, 22 and 36 and 41.
3-In section 2. Materials and Methods, calculation of mole fractions should be defined clearly.
Answer:
I think this is a very simple principle and very well-known calculations.
4-The standard method for testing microhardness should be provided.
Answer:
New reference no. 38 was added.
5-Authors claim reaction 1 was the only reaction could be possibly happened during synthesis. All possible reactions should be provided and compared to each other by thermodynamic software such as HSC.
Answer:
This is not the general method for discussion. Giving some details about all the expected reactions is usually used in case of elucidating the mechanism which is not a target of this work.
6-In Fig.3, some peaks were not identified.
Answer:
The peaks are completely identified in case of the sample without additions. Only the new appearing phase (ZrO2) is identified in the other XRD diagrams. No need for repeating the symbols of the other phases.
7-Unlike Fig.5, it seems that the sample was not fully-dense after the synthesis process based on Fig.6. why?
Answer:
Fig. 6 represents a very high dense product and not a fully dense product. The data presented in Fig. 5 related to the open porosity but there still are some closed pores.
8-The intrinsic value of hardness for each phase (ZrN,ZrSi2,ZrO2) should be provided to discuss the hardness of the composite by increasing ZrO2 content.
Answer:
Actually the comparison will not introduce a right figure, because the product is a composite of three phases and each phase has a different fraction. In addition the bonding between the particles of the different phases will be totally different from the bonding in case of single phase sample. The comparison will be very difficult.
9-More tests and data are needed to evaluate mechanical properties.
Answer:
This product was designed to work as abrasive material. The sample hardness is the most important property which could determine the success or failure of the sample.
10-The Conclusion section needs to be rewritten
Answer:
The conclusion section was modified
Reviewer 3 Report
The paper presents the results of studying the properties of composite materials based on ZrN / ZrSi2 doped with ZrO2, using XRD, SEM, EDS methods, measurements of porosity, density and hardness. This line of research is of scientific interest and may be of interest to a wide range of readers and researchers involved in such experiments. This work is of scientific interest, and also fully corresponds to the subject matter of the declared journal. In my opinion, this article can be accepted for publication after the authors answer the following questions.
1 The authors should explain exactly how the values ​​of density and porosity of composites were determined.
2 What is the reason for the drop in hardness at a high concentration of ZrO2 in the composition of composites?
3 In the presented X-ray diffraction patterns, the peaks characteristic of the ZrO2 phase are clearly visible, however, the authors do not provide any data on its content, as well as the phase concentration depending on the dopant content.
4 Have the authors established substitutional solid solutions, or did the low concentration of dopants not lead to the formation of complex phases?
5 SEM images require additional explanations, whether the porosity of the obtained samples was estimated by analyzing the obtained images or determined by the method of X-ray diffraction.
Author Response
Reviewer 3
1-The authors should explain exactly how the values ​​of density and porosity of composites were determined.
Answer: The method is clearly reported in the experimental part with two references for the exact steps of the methods. Ref 38 is the ASTM standard for determining the sample porosity and density that explain the details of the test.
Line 121 123. “The samples’ porosity and apparent density were measured by Archimedes’ method [38,39]. Archimedes’ principle is based on saturating the pores of the sample with water by immersing the samples under water and boiled for 2 h.”
2 What is the reason for the drop in hardness at a high concentration of ZrO2 in the composition of composites?
Answer: The reason for the drop in hardness at a high concentration of ZrO2 was discussed in the text (line306 to line 310). However additional detailed was added as follows:
Original text:” The drop of the hardness at 30 wt% ZrO2 could be correlated with the sudden increasing in the sample porosity, Figure 7. Both the increased sample porosity and the decreased value of hardness were arisen from the decrease in the adiabatic temperature and the liquid phase content as predicted by the thermodynamic calculations, Figure 1”
Modified text: The drop of the hardness at 30 wt% ZrO2 could be correlated with the sudden increasing in the sample porosity, Figure 7. Both the increased sample porosity and the decreased value of hardness were arisen from the decrease in the adiabatic temperature and the liquid phase content as predicted by the thermodynamic calculations, Figure 1. The liquid phase is responsible for facilitating the sintering of the sample during compression. The decrease in the fraction of liquid phase will have bad impact on the sintering of the sample which produces a sample with higher porosities.
3 In the presented X-ray diffraction patterns, the peaks characteristic of the ZrO2 phase are clearly visible, however, the authors do not provide any data on its content, as well as the phase concentration depending on the dopant content.
Answer: The different dopant contents were recorded on the XRD diagrams, Fig. 3. The dopant did not participate in the reaction and did not form side byproducts. It works only as a reinforcement of the ceramic matrix and improve its hardness.
4 Have the authors established substitutional solid solutions, or did the low concentration of dopants not lead to the formation of complex phases?
Answer: Actually neither solid solution nor complex phases were formed during the process. The XRD showed that dopant ZrO2 diffraction peaks are clearly visible which means that it does not participate in any side reactions.
5 SEM images require additional explanations, whether the porosity of the obtained samples was estimated by analyzing the obtained images or determined by the method of X-ray diffraction.
Answer: The sample porosity was determined only by Archimedes’ method as stated in the experimental part (line121-122). The image analysis and XRD did not used to determine the porosity in this work.
Round 2
Reviewer 1 Report
The authors took note of my comments and revised the manuscript. I recommend the manuscript to accept for publication.
Author Response
Comment: English language and style are fine/minor spell check required
Answer: The language has been completely revised
Reviewer 2 Report
Authors had a chance to modify the revised manuscript. Unfortunately, the paper still contains serious issues that prevents me from suggesting its publication.
Author Response
The comments are very general and in some cases have no relation to the scientific method of revision. I was asked to explain how I calculated the molar ratio, for example.
He requested that all manuscript sections should be improved in his second round revision.
I am sorry I cannot do that.